# Sero-prevalence and associated risk factors of bovine brucellosis in Sendafa, Oromia Special Zone surrounding Addis Ababa, Ethiopia

Hadji Bifo[1], Getachew Gugsa[2]*, Tsegabirhan Kifleyohannes[1], Engidaw Abebe[2‡], Meselu Ahmed[2‡]

1 Department of Veterinary Medicine, College of Veterinary Sciences, Mekelle University, Mekelle, Ethiopia,
2 Department of Veterinary Medicine, School of Veterinary Medicine, Wollo University, Dessie, Ethiopia

☯ These authors contributed equally to this work.
‡ These authors also contributed equally to this work.
* gugsag@yahoo.com

## Abstract

Bovine brucellosis is an infectious bacterial disease caused by members of genus Brucella, affecting both animals and humans, and resulting in a serious economic loss in animal production sector and deterioration of public health. A cross-sectional study was conducted from November 2014 to April 2015 to determine the seroprevalence and associated risk factors of bovine brucellosis in Sendafa, Oromia Special Zone, Ethiopia. A total of 503 blood samples were collected using a simple random sampling technique from dairy cattle of above 6 months of age with no history of previous vaccination against brucellosis. All sera samples were subjected to both Rose Bengal Plate Test for screening and Complement Fixation Test for confirmation. Accordingly, the overall seroprevalence of bovine brucellosis in the study area was 0.40%. The result showed that the seroprevalence of bovine brucellosis in the study area was not statistically significant with all proposed risk factors. No reactors were observed in male animals. The seroprevalence was observed in animals without previous history of abortion. Moreover, information was gathered on individual animal and farm-level risk factors and other farm characteristics using a questionnaire. Awareness among society was poor, so the positive animals can be a potential hazard to animals and humans in the study area. Therefore, public education should be done to improve the awareness of the community on bovine brucellosis and its public health impact with due consideration on the safe consumption of food of animal origin.

## Introduction

Ethiopia is claimed to have the largest livestock population in Africa. The total cattle population of the country is estimated to be around 60.39 million. Out of this total cattle population in the country, 98.24%, 1.54%, 0.22% are local, hybrid, and exotic breeds, respectively. The female and male cattle constitute about 54.68% and 45.32%, respectively [1]. Despite the country has huge livestock resources, the production and productivity of the sector remain low due

**Data Availability Statement:** All relevant data are within the manuscript file.

**Funding:** The author(s) received no specific funding for this work.

**Competing interests:** The authors have declared that no competing interests exist.

to rampant infectious and parasitic diseases, feed shortage and malnutrition, poor and traditional management system, lack of infrastructure and veterinary service provision, and limited and unimproved genetic potential [2, 3]. Different infectious diseases of multiple etiologies may infect cattle and other animals both in developed and developing countries of the world [4].

Among infectious diseases, brucellosis is a major constraint for animal production which is a highly contagious, zoonotic, and economically important bacterial disease of animals worldwide with a great burden in developing countries [5]. The disease affects domestic animals (cattle, sheep, goat, camel, pig, and dogs), human, wildlife, and marine mammals [6]. This infectious zoonotic bacterial disease is caused by a member of the genus Brucella [7]. The genus Brucella is Gram-negative, facultative intracellular, coccobacillus, non- spore-forming, and non-motile bacteria comprised of different species affecting preferred host species [5]. Currently, ten species including the better-known six classical species comprised of *B. abortus*, *B. melitensis*, *B. suis*, *B. ovis*, *B. canis*, and *B. neotomae* are known. In recent times, other new species of the genus including *B. ceti*, *B. pinnipedialis*, *B. microti*, and *B. inopinata* which affect different species of animals are also identified [8].

Bovine brucellosis is the most important disease among other brucellosis affecting different animals in many countries due to its high economic importance [7]. It is a major zoonotic disease widely distributed in both humans and animals especially in the developing world [9]. It is caused principally by *B. abortus* and occasionally by *B. melitensis* and *B. suis* [10]. The epidemiology of the disease is complex and influenced by several factors including transmission and spreading of the disease [11]. Aborted fetuses, fetal membranes, vaginal discharges, and milk from infected cows are the main sources of infection [12]. The mode of transmission among animals is through the exposure of mucous membranes, direct contact with infected materials, or inhalation of aerosols [13]. According to Annapurna et al. [14], the main sources of Brucella infection in humans are occupational contact and consumption of contaminated foods of animal origin. Human-to-human transmission through tissue transplantation or sexual contact has also been reported [15].

Bovine brucellosis is characterized by abortion with retention of the placenta, metritis, weak calves, stillbirth, infertility, and reduced milk yield [10]. Infected bulls may show signs of infection including orchitis, and epididymitis [16, 17]. In humans, the disease is characterized by fever, depression, sweating, malaise, weight loss, joint pains, headache, and anorexia [18].

The economic and public health impact of brucellosis remains of particular concern in developing countries of the world mainly in the dairy production sector [18, 19]. The extensive economic losses of the disease are considered both in terms of animal production loss and public health [20]. In addition to its production loss, the disease impedes free animal movement and is a barrier to import and export livestock trade [18].

Though the information on how and when brucellosis was introduced and established in Ethiopia is not documented [21], bovine brucellosis was first reported in 1970 in the country [22]. Several serological studies done in different localities of Ethiopia indicated that bovine brucellosis is a widespread and endemic disease of cattle in different farming and production systems of the country. Though different studies have been conducted in different parts of Ethiopia, the disease is still a major problem demanding much research and investigation. Moreover, Sendafa is known by its high dairy production sector through the introduction of high potential crossbreed dairy cattle, but the status of bovine brucellosis in the area is still not well known. Thus, it is necessary to assess the status of bovine brucellosis in smallholder dairy farms in the area. Hence, the objectives of this study were to estimate the seroprevalence of bovine brucellosis and identify potential risk factors that could precipitate its occurrence in smallholder dairy farms in Sendafa, Oromia regional state of Ethiopia.

## Materials and methods

### Ethics approval and consent to participate

This study was reviewed and approved by the Research Ethics Committee of the College of Veterinary Sciences, Mekelle University. The study participants were informed about the study purpose and answered the questionnaire anonymously; they were free to skip any item they did not wish to answer.

### Study area

The study was conducted in Sendafa which is one of the administrative units of Oromia Regional Government special administration zone surrounding Addis Ababa. The area is situated in Berehna Aletu District, Northern Shewa, Oromia Regional State at a distance of 39 km to the north of Addis Ababa, the capital city of Ethiopia. The name Sendafa is taken from the Oromo name for a kind of thick, jointed grass or reed which grows in swampy areas. Astronomically, the town is located in the geographic coordinates between 9˚06'14" and 9˚10'30" North latitudes and 38˚57'60" and 39˚04'53" East longitudes with an elevation of 2514 meters above sea level. The area receives a mean annual rainfall of 1200mm in bimodal distribution (June to August and January to April) with the average temperature ranges from 15˚C to 24˚C. A mixed livestock production system with crop farming is practiced in the area [23, 24]. The map of the study area that was generated by us is shown below in Fig 1.

### Study design and population

A cross-sectional study was conducted from November 2014 to April 2015 to estimate the seroprevalence of bovine brucellosis and to identify the potential risk factors associated with the occurrence of the disease in the study area. Study animals were selected from 20 different dairy farms that are found in the study area. Animals included for this study were cross breed dairy cattle comprised of both sexes, above 1 year of age, managed under intensive management system, and with no vaccination history against brucellosis. Besides, a total of 20 farm owners and workers of both sexes (10 males and 10 females) were interviewed.

### Sample size determination

The sample size (n) required for the study was estimated using the statistical formula given by Thrusfield [25].

$$n = \frac{z^2 \, P_{exp}(1-P_{exp})}{d^2} \quad n = \frac{1.96^2 \, P_{exp} \, (1-P_{exp})}{d^2}$$

Where, n = sample size, z = statistic for a level of confidence
d = required absolute precision, $P_{exp}$ = expected prevalence

For the calculation, 95% confidence interval (z), 5% absolute precision (d), and 50% expected prevalence (P) of bovine brucellosis, since the magnitude of bovine brucellosis in Sendafa was not known, were used. Based on the above formula the minimum desired sample size was calculated to be 384, but to increase the precision of the study the sample size was increased to 503.

### Sampling technique, sample collection, and processing

Simple random sampling technique was employed to select samples from smallholder dairy farms in the study area. About 5–10 ml of blood samples were collected from each sampled

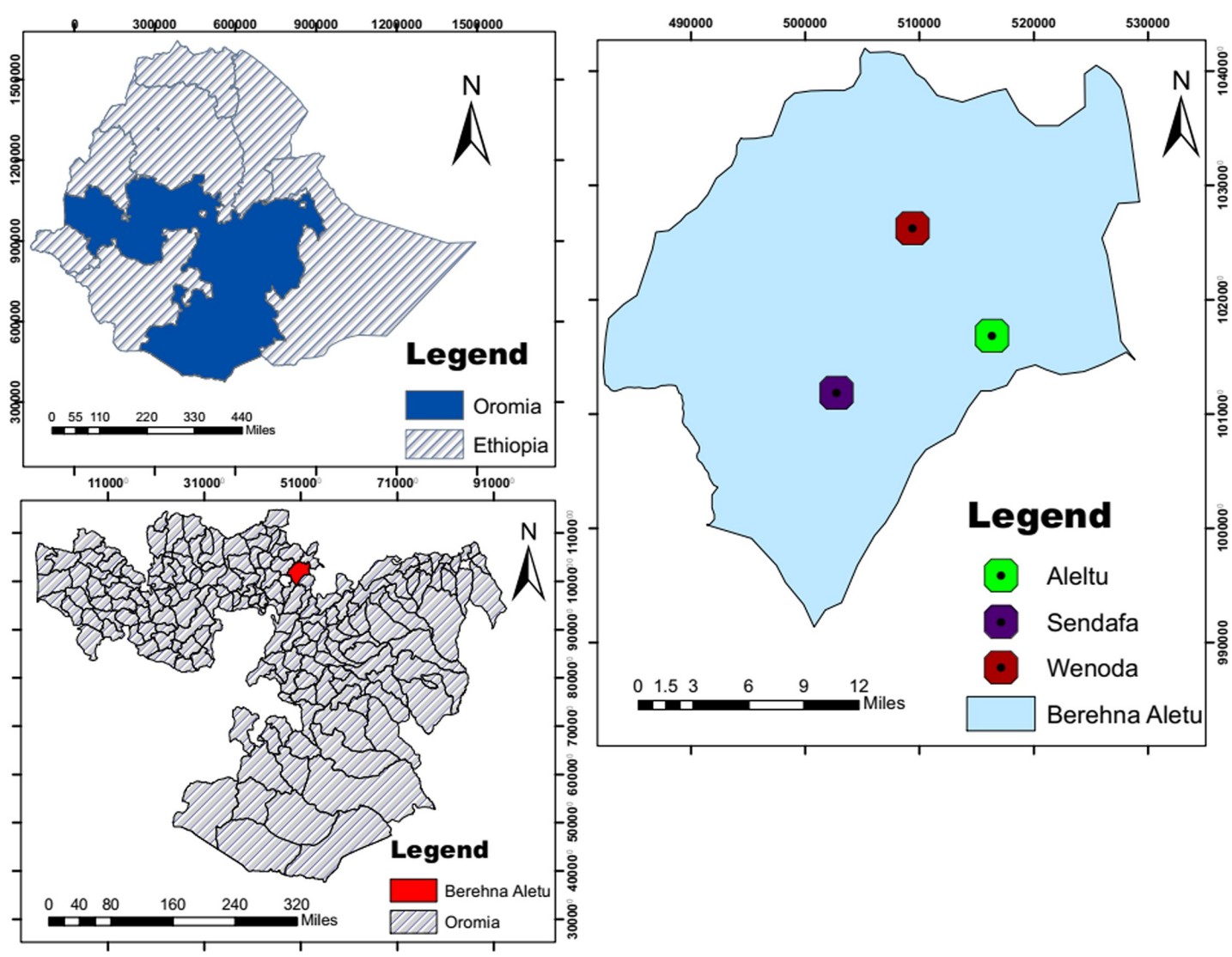

**Fig 1. Map of the study area, Sendafa.**

animal via jugular-vein puncture using plain vacutainer tubes and needles after proper restraining and disinfecting of the site using 70% alcohol. Each sample was labeled with specific standard identification and transported to Asella Regional Laboratory in an icebox for further processing and Rose Bengal Plate Test (RBPT). The tubes were set tilted overnight at room temperature to allow clotting and in the next morning, sera were harvested from the clotted blood by sterile and labeled cryovial tubes. All the sera were kept at -20˚C till the RBPT was performed and later transported to National Veterinary Institute (NVI) Laboratory, Debre-Zeit, Ethiopia, with icebox for confirmation and the samples were stored at -20˚C until processed.

**Serological test procedures.** *Rose Bengal Pate Test (RBPT)*. All collected sera samples were screened using the Rose Bengal Plate Test in Asella Regional Laboratory as per the procedure described by Alton et al. [26] and OIE [27]. The results were read by examining agglutination and the degree of agglutination was recorded as 0 (indicates the absence of agglutination), + (indicates barely visible agglutination), ++ (indicates fine agglutination) +++

(indicates coarse clumping). Those samples identified with no agglutination (0) were recorded as negative while those with +, ++ and +++ were recorded as positive [28].

*Complement Fixation Test (CFT).* The two sera samples which tested positive to RBPT were further confirmed using the complement fixation test at the National Veterinary Institute, Department of Immunology, Debre-Zeit, Ethiopia according to the protocols recommended by OIE [5]. Antigen, control sera, and complement were obtained from the Federal Institute for Consumer Health Protection and Veterinary Medicine (BgVV), Berlin, Germany. The preparation of sheep red blood cells (SRBC), the methods of CFT test, and preparation of reagents were according to the protocol of the BgVV Service Laboratory. Positive reactions were indicated by sedimentation of sheep red blood cells (SRBC) and the absence of hemolysis while negative reactions were revealed by hemolysis of SRBC. Sera with a strong reaction, more than 75% fixation of complement at a dilution of 1:10, and at least with 50% fixation of complement at a working dilution (1:5) was interpreted as a positive result [29].

## Questionnaire survey

A standard structured questionnaire was prepared to obtain general information on the potential risk factors associated with the occurrence of bovine brucellosis in the study area. For this, verbal consent was obtained from twenty respondents, from which their animals tested for brucellosis, and the objective of the survey explained to them before the start of the interview. Then, the questionnaire was administered for those selected individuals and the required information was collected. The questionnaire focused on the source of dairy cattle, awareness of the respondents on the risk of transmission of brucellosis from cattle to cattle and cattle to human, management practices, dead-animal(s), and aborted material disposal practices, handling of retained placenta and consumption of raw milk.

## Data management and analysis

All raw data that were collected from laboratory investigations and questionnaire survey were compiled and summarized. The coded data were entered into Microsoft Excel 2007 spreadsheet and transferred to STATA Version 11 for statistical analysis. Descriptive and analytic statistics were employed and the Chi-square test ($\chi^2$) was computed to see the association of proposed risk factors with that of the occurrence bovine brucellosis. The statistically significant association between variables and the disease was considered when the P-value was less than 0.05.

## Results

The overall seroprevalence of bovine brucellosis in the study areas was 0.40% which was recorded on the bases of both RBPT and CFT (Table 1). In addition, different expected potential variables were considered to assess their contribution to the occurrence of bovine brucellosis as illustrated in Table 2.

The difference in the seroprevalence of bovine brucellosis among the different age groups was not statistically significant (P>0.05). The recorded seroprevalence of the disease in the

**Table 1. The overall seroprevalence of bovine brucellosis in the study area.**

| Serological tests | Total No. of animals tested | Total No. (%) of positive animals |
|---|---|---|
| RBPT | 503 | 2 (0.40) |
| CFT | 2 | 2 (100) |
| Total | 503 | 2 (0.40) |

**Table 2. Sero-prevalence of bovine brucellosis among different risk factors.**

| Variable | No. of tested animals | No. of positive (%) | $\chi^2$ | P-value |
|---|---|---|---|---|
| **Age** | | | 1.3626 | 0.506 |
| Young | 87 | 0 (0.00) | | |
| Adult | 318 | 1 (0.31) | | |
| Heifer | 98 | 1(1.02) | | |
| **Sex** | | | 0.0283 | 0.866 |
| Male | 7 | 0 (0.00) | | |
| Female | 496 | 2 (0.40) | | |
| **Origin** | | | 0.3464 | 0.556 |
| Born | 74 | 0 (0.00) | | |
| Bought | 429 | 2 (0.47) | | |
| **Breeding method** | | | 4.5082 | 0.105 |
| Natural mating | 208 | 0 (0.00) | | |
| AI | 140 | 0 (0.00) | | |
| Both | 155 | 2 (1.29) | | |
| **History of abortion** | | | 0.0080 | 0.0929 |
| Aborted | 2 | 0 (0.00) | | |
| Non aborted | 501 | 2 (0.40) | | |
| **Overall** | **503** | **2 (0.40)** | | |

young, adults, and heifers was found to be 0.00%, 0.31%, and 1.02%, respectively as listed in Table 2.

The sexes of the tested animals didn't seem to have a significant association (P>0.05) with the seroprevalence of bovine brucellosis. The recorded seroprevalence of the disease in the current study was 0.40% and 0.00% in female and male dairy cattle, respectively (Table 2).

Similarly, there was no significant association between the origin (source) of animals and the seroprevalence of bovine brucellosis (P>0.05). The seroprevalence of bovine brucellosis in home born and bought dairy cattle was 0.00% and 0.47%, respectively as stated in Table 2.

The seroprevalence of bovine brucellosis in different dairy cattle that use natural mating, artificial insemination (AI), and both methods of breeding were found to be 0.00%, 0.00%, and 1.29%, respectively. Thus, the method of breeding didn't have a significant association with the seroprevalence of bovine brucellosis (P>0.05) as illustrated in Table 2.

Moreover, the history of abortion in the dairy cattle hadn't a significant association with the prevalence of bovine brucellosis (P>0.05). The seroprevalence of bovine brucellosis in dairy cattle with a history of abortion and not aborted were found to be 0.00% and 0.40%, respectively (Table 2).

The questionnaires were administered to 20 intensive farm owners and workers comprised of equal proportion of both sexes. Forty-five percent of them were attained their primary school and 55% of them were attained secondary school and above. Of the total respondents, 35% were used AI breeding method, 30% were used natural mating and 35% were used both breeding systems. Ninety percent of them had open housing system and used proper disposal of materials after birth; and 65% of them limit their animals from having contact with other animals rather than their farms as illustrated in Table 3.

## Discussion

The current serological study revealed that the overall prevalence of bovine brucellosis in the study area was 0.40%. This finding was in agreement with the findings of Bashahun et al. [30]

**Table 3. Status of farm owners and their farm management systems obtained from the questionnaire survey.**

| Variable | Category | Total No. (%) of Respondents |
|---|---|---|
| **Sex** | Female | 10/20(50%) |
| | Male | 10/20(50%) |
| **Age** | Adult | 20/20(100.00%) |
| | Young | 0/20(0.00%) |
| **Educational level** | Primary | 9/20(45.0%) |
| | Secondary and above | 11/20(55.0%) |
| **Breeding method** | AI | 7/20(35%) |
| | Natural | 6/20(30%) |
| | Both | 7/20(35%) |
| **House type** | Opened | 18/20(90%) |
| | Closed | 2/20(10%) |
| **Proper disposal after birth** | Yes | 18/20(90%) |
| | No | 2/20(10%) |
| **Limit contacts to exposure** | Yes | 13/20(65%) |
| | No | 7/20(35%) |
| **Sex of animals** | All female | 15/20(75%) |
| | All male | 0/20(0.00%) |
| | Mixed | 5/20(25%) |

(0.3%), Bashitu et al. [31] (0.2%), Tadele [32] (0.61%), Yayeh [33] (0.14%), Tolosa [34] (0.77%), Pal et al. [35] (0.78%), who reported in selected districts of Jimma zone, Debrebirhan and Ambo Towns, Jimma, North Gondar Zone, Southwestern Jimma zone and North Shewa, respectively. However, the present finding was higher than the finding of Bedaso et al. [36] (0.06%) in Addis Ababa.

In the other hand, the present finding was lower than the previous research works conducted in different parts of Ethiopia by Abay [37] in Arsi, Gebawo et al. [38] in Adami Tullu, Hunduma and Regassa [21] in East Shoa Zone, Hailu et al. [39] in Jig-Jiga zone of Somali Regional State, Gebreyohans [40] in Addis Ababa, Tariku [41] in Chafa State Dairy Farm, Taye [42] in Abernosa Cattle Breeding Ranch, and Mussie et al. (2007) [43] in Bahir Dar milk shed who reported magnitude of 4.9%, 4.3%, 4.1%, 1.38%, 1.5%, 22%, 19.5%, and 4.63%, respectively. Moreover, the present finding was by far much lower than the findings of Wossene et al. [44] (14.6%), Eyob et al. [45] (9.87%), Mekonnen et al. [46] (6.1%), Alehegn et al. [47] (4.9%), Gelma et al. [48] (4.7%), Hika et al. [49] (3.75%), Berhe et al. [50] (3.19%), Megersa et al. [51] (3.5%), Kemal and Minda [52] (4.95%), Jergefa et al. [22] (2.9%), Hagos et al. [53] (2.4%), Fekadu et al. [54] (2%), Dinknesh et al. [55] (1.04%), Moti et al. [56] (1.97%), Yohannes et al. [57] (2.6%), Abera et al. [58] (2.7%), Bulcha et al. [59] (1.04%) and Yitagele et al. [60] (1.3%) who reported in Jikow District (Gambella), Asella, Western Tigray, Gondar Town, Borana Zone, Bishoftu Town, Tigray, Southern and Eastern Ethiopia, Agarfa and Berbere Districts of Bale Zone, Central Oromiya, Alage district, Eastern Showa, Becho District, Guto-Gida district, Arsi Zone, Hawassa Town, Adama Town and Eastern Ethiopia, respectively.

The difference in the seroprevalence of bovine brucellosis among the different reports from different areas of the country might be due to the agro geographical difference, difference in management and husbandry practices, source of replacement animals, educational status of farmers, hygienic practice in the farms, and availability of maternity pens at calving which decreases the exposure of infected and susceptible animals [61].

In the current study, although the difference in seroprevalence between the two sexes was not statistically significant, there was no positive reactor among male animals. This finding

was in agreement with the findings of Tadele [32] (0.00% and 0.97%), Bashitu et al. [31] (0.00% and 0.2%), Dinknesh et al. [55] (0.00% and 3.13%), Gebawo et al. [38] (0.00% and 3.1%), and Bashahun *et al.* [30] (0.00% and 1.8%) who reported only female positive animals. The absence of positive male animals in the current study might be due to the smaller number of male animals examined as compared to females or it might be due to the justification given by Kebede et al. [62] who stated that male animals are less susceptible to *Brucella* infection due to the low level of erythritol.

In this study, despite there was no significant association between the age categories of the tested animals and seroprevalence of bovine brucellosis, infected animals were adult and mature heifer. This finding was consistent with the findings of Tadele [32] who reported 0.00% and 1.2% in young and adult cattle, respectively; Bulcha et al. [59] who reported 0.00% and 1.27% in young and adult cattle, respectively and Nuraddis et al. [63] who reported 2.38% and 4.32% in young and adult cattle, respectively. The comparative high occurrence of bovine brucellosis in adult animals could be due to sexual maturity which is a very important condition for the rapid multiplication of *Brucella* organism [64–66]. Thus, sexually mature and pregnant cattle are more susceptible to *Brucella* infection as compared to sexually immature animals [67]. Moreover, according to Radostits et al. [61], younger animals tend to be more resistant to infection and frequently clear infections through latent infection could occur.

In the present study, the origin history of animals (born or bought) didn't show significant association with the occurrence of bovine brucellosis. However, positive reactors were found in animals with purchase history. These animals might be purchased from farms infected with bovine brucellosis. This indicates outside sources for stock replacement could be one possible way of the introduction of the disease into unaffected farms.

In the current study, the method of breeding didn't have a significant association with the seroprevalence of bovine brucellosis. However, the use of both AI and natural service method in the farms were found to be sources for *Brucella* infection in this study. The purchase of infected bulls or contamination of frozen semen with *Brucella* could not be ruled out [68].

According to the current study, dairy cows without abortion history were detected positive for brucellosis though history of abortion in the dairy cattle hadn't significant association with the prevalence of bovine brucellosis. Cows with a history of abortion were found to be zero. In contrast to this report, Hika et al. [49] (2.82% and 14.63%), Dinknesh et al. [55] (0.00% and 17.4%) and Bulcha et al. [59] (0.00% and 19.05%) reported a significantly higher prevalence of bovine brucellosis in cows with abortion history. Animals included with the history of abortion in this study might be aborted due to other causes. This lack of association between history of abortion and seroprevalence of bovine brucellosis suggests that other causes largely outweigh brucellosis as a cause of abortion and stillbirth [10, 61, 69, 70]. In addition, the abortion rate in infected animals is dependent on many factors and varies according to the period for which the cows have been infected, management practices, the susceptibility of the pregnant females, and other various environmental factors [13].

## Conclusion and recommendations

The present study indicated that the occurrence of bovine brucellosis in Sendafa dairy farms is at a low magnitude. Even though the seroprevalence is low, it can still be a potential hazard for both susceptible animals and humans as the awareness among the society was poor in the study area. All proposed risk factors including age, sex, history of abortion, and breeding method in the study site showed insignificant variation. Test and slaughter program is not possible in countries like Ethiopia where compensation cannot be made for slaughtered animals. Hence, alternative control measures that are feasible and acceptable under local conditions

have to be designed and well implemented. Coordinated surveillance and monitoring system for bovine brucellosis should be carried out to design appropriate and effective control and prevention strategies against the disease in the study area at large in the country. Animal intensification should be followed by efficient and effective disease control programs. Moreover, public awareness on economic as well as public health impacts of bovine brucellosis should be created.

## Acknowledgments

We would like to express our sincere gratitude to Asella Regional Veterinary Laboratory Staff members particularly Showil Kebede, and Sendafa farm owners and farm workers for their voluntariness, contribution, and guidance during the research work.

## Author Contributions

**Conceptualization:** Hadji Bifo, Getachew Gugsa, Tsegabirhan Kifleyohannes.

**Data curation:** Hadji Bifo, Getachew Gugsa.

**Formal analysis:** Hadji Bifo, Getachew Gugsa.

**Investigation:** Hadji Bifo, Getachew Gugsa.

**Methodology:** Hadji Bifo, Getachew Gugsa, Tsegabirhan Kifleyohannes, Engidaw Abebe, Meselu Ahmed.

**Supervision:** Getachew Gugsa.

**Writing – original draft:** Hadji Bifo, Getachew Gugsa, Tsegabirhan Kifleyohannes.

**Writing – review & editing:** Hadji Bifo, Getachew Gugsa, Engidaw Abebe, Meselu Ahmed.

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
