## [Decision Letter · Decision Letter 0]

16 Sep 2020

PONE-D-20-24877

Sero-Prevalence and Associated Risk Factors of Bovine Brucellosis in Sendafa, Oromia Special Zone Surrounding Addis Ababa, Ethiopia

PLOS ONE

Dear Dr. Gugsa,

Thank you for submitting your manuscript to PLOS ONE. After careful consideration, we feel that it has merit but does not fully meet PLOS ONE’s publication criteria as it currently stands. Therefore, we invite you to submit a revised version of the manuscript that addresses the points raised during the review process.

Many thanks for submitting your manuscript to PLOS One

It was reviewed by two experts in the field, and they have requested some changes be made prior to acceptance.

If you could make these changes and write a response to reviewers, that will greatly expedite revision upon resubmission

I wish you the best of luck with your changes

Hope you are keeping safe and well in these difficult times

Thanks

Simon

We look forward to receiving your revised manuscript.

Kind regards,

Simon Clegg, PhD

Academic Editor

PLOS ONE

2. In your Methods section, please provide additional details regarding the animals used in your study and ensure you have described the source. For more information regarding PLOS' policy on materials sharing and reporting, see https://journals.plos.org/plosone/s/materials-and-software-sharing#loc-sharing-materials.

Reviewers' comments:

Reviewer's Responses to Questions

**Comments to the Author**

1. Is the manuscript technically sound, and do the data support the conclusions?

Reviewer #1: Yes

Reviewer #2: Yes

2. Has the statistical analysis been performed appropriately and rigorously? 

Reviewer #1: I Don't Know

Reviewer #2: Yes

3. Have the authors made all data underlying the findings in their manuscript fully available?

Reviewer #1: Yes

Reviewer #2: Yes

4. Is the manuscript presented in an intelligible fashion and written in standard English?

Reviewer #1: Yes

Reviewer #2: No

5. Review Comments to the Author

Reviewer #1: The manuscript presents an original work and the study is related to an important zoonotic disease. However, there is a need to improve it in several aspects: the author need to clarify the herd prevalence, within herd prevalence and overall seroprevalence; there are some grammatical errors and should be revised for the improvement;

I recommend being accepted with several revision

Reviewer #2: Interesting paper that highlights a potential hazard of a zoonotic disease. The estimation of a mouch lower prevalence than other studies is of some interest and may need further investigation in order to evaluate the sampling procedure of other studies

---

## [Author Response · Author response to Decision Letter 0]

11 Oct 2020

First of all, we would like to say both Reviewers many thanks for their invaluable time and critical and professional evaluation of our manuscript. We have a great appreciation for the corrections given by the reviewers and found them more valuable for the betterment of our manuscript.

When I come to the comments were given by the reviewers’:

Methods Section:

1. The comment given by the reviewers related to the inclusion/exclusion criteria of the sampled animals is quite right. We didn’t take any sample from animals below 1 year of age. Hence, I have made the correction as per the comment.

2. Of course, as a professional we had to include the national/regional control program of brucellosis. However, so far there is no any proclamation related to the control program of this disease at regional as well as national level. That is way we didn’t say anything related to it. But, we have indicated it in the conclusion and recommendation section of this manuscript. 

3. We didn’t come across for declaring a positive herd as well a positive animal. Moreover, as the title of the manuscript as well as the methodological approach that we used, we didn’t perform any bacteriological as well as molecular characterization techniques.

4. Related to the herd level and within the herd prevalence’s calculation, though as a professional we know such calculations are very crucial, however, we didn’t perform that. Since our country, Ethiopia, didn’t apply the testing and replacement and/or slaughter policies. Moreover, from the beginning of such a research here in Ethiopia, the farm managers and/or owners didn’t want to be notified regarding the status of their farms as well as the herd prevalence of each farm of the study area.

Result Section:

1. I have included map of study area as well as the region as per the comments.

---

## [Editor Report · Decision Letter 1]

30 Oct 2020

Sero-Prevalence and Associated Risk Factors of Bovine Brucellosis in Sendafa, Oromia Special Zone Surrounding Addis Ababa, Ethiopia

PONE-D-20-24877R1

Dear Dr. Gugsa,

We’re pleased to inform you that your manuscript has been judged scientifically suitable for publication and will be formally accepted for publication once it meets all outstanding technical requirements.

Kind regards,

Simon Clegg, PhD

Academic Editor

PLOS ONE

Additional Editor Comments:

Many thanks for resubmitting your manuscript to PLOS One

As all comments have been addressed and the manuscript reads well, I have recommended your manuscript for publication

You should hear from the Editorial Office soon

It was a pleasure working with you and I wish you all the best for your future research

Hope you are keeping safe and well in these difficult times

Thanks

Simon

---

## [Editor Report · Acceptance letter]

6 Nov 2020

PONE-D-20-24877R1 

Sero-Prevalence and Associated Risk Factors of Bovine Brucellosis in Sendafa, Oromia Special Zone Surrounding Addis Ababa, Ethiopia 

Dear Dr. Gugsa:

I'm pleased to inform you that your manuscript has been deemed suitable for publication in PLOS ONE. Congratulations! Your manuscript is now with our production department. 

Kind regards, 

on behalf of

Dr. Simon Clegg 

Academic Editor

PLOS ONE